# Changes in Soil Humin Macromolecular Structure Resulting from Long-Term Catch Cropping

**DOI:** 10.3390/molecules29215049

**Published:** 2024-10-25

**Authors:** Jerzy Weber, Elżbieta Jamroz, Lilla Mielnik, Riccardo Spaccini, Andrzej Kocowicz, Irmina Ćwieląg-Piasecka, Maria Jerzykiewicz, Danuta Parylak, Magdalena Dębicka

**Affiliations:** 1Institute of Soil Science, Plant Nutrition and Environmental Protection, Wroclaw University of Environmental and Life Sciences, Grunwaldzka 53, 50-375 Wrocław, Poland; elzbieta.jamroz@upwr.edu.pl (E.J.); andrzej.kocowicz@upwr.edu.pl (A.K.); irmina.cwielag-piasecka@upwr.edu.pl (I.Ć.-P.);; 2Department of Bioengineering, West Pomeranian University of Technology in Szczecin, Papieża Pawła VI 3, 71-459 Szczecin, Poland; lilla.mielnik@zut.edu.pl; 3Research Center CERMANU, University of Naples, Piazza Carlo di Borbone, 80055 Portici, Italy; riccardo.spaccini@unina.it; 4Faculty of Chemistry, University of Wroclaw, 50-137 Wrocław, Poland; maria.jerzykiewicz@chem.uni.wroc.pl; 5Institute of Agroecology and Plant Production, Wroclaw University of Environmental and Life Sciences, Grunwaldzki Sq. 24A, 50-363 Wrocław, Poland; danuta.parylak@upwr.edu.pl

**Keywords:** long-term field experiment, fluorescence, UV-Vis, NMR, EPR, humic substances

## Abstract

The aim of this study was to assess the effect of long-term catch crop application on the structural properties of humin, which is considered the most recalcitrant fraction of soil organic matter. Soil samples from a 30-year field experiment on triticale cultivated with and without catch crops were analysed to determine the total organic carbon content and fractional composition of humic substances. Meanwhile, humin isolated from bulk soil was analysed to determine its elemental composition and spectroscopic properties measured with UV-Vis, fluorescence, and ^13^C-CPMAS-NMR. It was found that catch crop farming enhanced the formation of highly reactive humus substances, like low-molecular-weight fractions and humic acids, while decreasing the humin fraction. The higher H/C and O/C atomic ratios of humin and the UV-Vis, fluorescence, and ^13^C-CPMAS-NMR results confirmed a higher share of oxygen-containing functional groups in humin isolated from the soil with catch crop rotation, also corroborating its greater aliphatic nature. Under the conditions of our field experiment, the results indicated that organic residues from catch crops quickly undergo the decay process and are transformed mainly into highly reactive humus substances, which can potentially improve soil health, while mineral fertilisation alone without catch crops favours the stabilisation and sequestration of carbon.

## 1. Introduction

Soil organic matter (SOM) is widely recognised as a major factor affecting soil fertility. SOM affects nutrient cycling and the biological, physical, and chemical properties of soil but also plays a significant role in the global C cycle [1,2,3,4,5]. The type of land use and crop management influences many soil properties, such as sorption and buffer functions and land productivity, as well as the content and quality of soil organic matter [6]. In many developed countries, the utilisation of organic matter as a source or nutrient elements has been replaced by inorganic fertilisers, which can lead to a significant change in soil properties and the depletion of SOM content as well as quality [7].

Most of the SOM constituents occur as humic substances (humic acids—HAs, fulvic acids—FAs, and humin—HUM), which are highly reactive and recalcitrant fractions [8,9,10,11]. These substances are characterised by different properties, especially the ability to form complexes with the mineral components of soil. FAs have a greater affinity to create these connections, the molecules of which have a simpler structure than HAs and are less condensed with more side chains. They also have greater solubility and, thus, a greater ability to move along the soil profile. What is more, they regulate the availability of nutrients for plants, modify the population of microorganisms inhabiting a given ecosystem, and affect the circulation of elements in nature and their flow between terrestrial and aquatic ecosystems. HAs and FAs, under the influence of external factors, can dissociate H^+^ ions from the functional groups, resulting in the appearance of negative charges that contribute to the cation exchange capacity of the soils [12]. The humification of organic matter is usually associated with a variation in the elemental composition of the molecules of humic substances, which is expressed by variations in the proportion of C, H, N and O, as well as modifications of the most valuable functional groups (mainly carboxylic and phenolic). More humified organic matter, compared to less humified, is distinguished by an enhancement in the molecules’ aromaticity, a decrease in the hydrogen content, an increase in the carbon content and an increased number of carboxyl and methoxy groups [13].

Until now, the humin fraction has been the least known constituent of SOM. However, recent reports indicate that this fraction may represent an important part of the carbon pool responsible for carbon sequestration and the binding of pollutants [14,15,16,17,18]. Using advanced instrumental techniques such as FTIR, EPR, ^13^C CP MAS NMR, spectroscopy, and fluorescence analysis enable the determination of the molecular composition of humic substances, indicating the direction of OM transformations as regards the precursor organic residues [19,20,21].

A catch crop is a fast-growing crop sown in fields between two main crops. Catch crops can be a valuable source of organic matter and plant nutrients, increasing the biological activity of the soil and improving its fertility. The fertilising value of catch crops well chosen for crop rotation is often compared to the fertilising value of manure. Catch crops increase the potential of agriculture to sequester carbon, which is particularly important in the era of the need to reduce the negative impact of this sector on the natural environment and fit the requirements of European policy [22]. Species that quickly produce green mass are used as catch crops and, after ploughing in, quickly decompose, enriching the soil with valuable nutrients [23]. The influence of catch crops on soil’s physical properties and the balance of basic macronutrients (N, P) is quite well known [24,25]. There are no studies in the literature on the influence of this type of cultivation on the quality of organic matter, in particular on the properties of the humin fraction, which is a fraction relevant for C sequestration.

Therefore, the aim of this research was to determine the impact of long-term catch crop application on the quality of SOM, particularly the chemical structure of humin, which is the most recalcitrant fraction of SOM.

## 2. Results and Discussion

### 2.1. Total Organic Carbon Content and Fractional Composition of Humus Substances

Over 30 years of using catch crops did not cause significant changes in the organic carbon content, but it significantly influenced the transformation of soil organic matter, contributing to changes in its fractional composition (Table 1). First of all, the soil cultivated in this way (H10) showed a significantly higher low-molecular-weight fraction (FF) and a higher amount of humic acids (HAs). On the other hand, the soil cultivated without catch crops (H9), compared with that cultivated with catch crops (H10), showed a significantly higher humin fraction (36.8%). This suggests that organic residues from catch crops quickly undergo the decay process and transform mainly into highly reactive humic substances, potentially improving soil health, while mineral fertilisation alone without catch crops rather favours the stabilisation and sequestration of carbon. This phenomenon was confirmed by the significant correlation coefficients for humus fractions calculated for soil without catch crops (Table 2): the higher the total extractable carbon, the lower the C content in the humin fraction.

### 2.2. Elemental Composition of Humin Fraction

The elemental composition of humin is presented in Table 3. The content of carbon in humin from the plots where catch crops were applied (H10) was lower (44.54%) than those without catch crops (50.75%), while their H, N, and O contents were higher, indicating their less aromatic structures [26,27]. Ash content was in the range of 9.76–18.18%. This is a good indicator of the effective procedure used during humin isolation [27,28,29]. The higher H/C and O/C atomic ratios of humin molecules that originated from the soil where catch crops were applied (H10) confirmed that their structure is more aliphatic [26,30] than that of the H9 sample.

### 2.3. UV-Vis Properties of Humin

The obtained absorption spectra are distinguished by the occurrence of a characteristic band in the 250–300 nm region with a maximum at a wavelength of 280 nm (Figure 1).

This signal is connected with the π-π* type electronic transition and is typical for the chromophores that are aromatic rings with different degrees and types of substitution [31]. The strong absorption in this area is attributed to the presence of aromatic alcohols and ketones, and it reflects the content of lignin-type compounds [32]. The sample of humin that originated from the soil where catch crops were applied (H10) showed higher absorption signals in this region compared to humin sample H9.

Higher values of the calculated coefficients for H10 (Table 4) pointed to its lower degree of condensation of aromatic structures and lower molecular weight compared to H9 [33,34]. Machado et al. [21] suggested that humin with high E_465_:E_665_ values originate from a relatively young material of a low humification degree.

According to Senesi et al. [35], absorbance at 280 nm proves the presence of material in the initial stage of organic matter transformation (lignin-like structures and quinone fragments). Therefore, the E_280_:E_465_ coefficient is used as an indicator of the presence of structures characterising various stages of humification [36], but also the amount of lignin in the isolated samples [37]. In turn, the higher value of the E_280_:E_465_ ratio in sample H10 suggests the regular addition of fresh biomass, which causes its enrichment in lignin, compared to H9. Furthermore, the higher value of the E_280_:E_400_ coefficient observed for H10 extracted from catch-crop-applied soil points to its relatively higher proportion of oxygen-containing functional groups, including carboxyl groups [38]. This confirms its greater aliphatic nature, as indicated by the results of elemental composition analysis (Table 3).

### 2.4. Fluorescence Properties

In the EEM spectra, three main regions can be distinguished, identified by a pair of excitation and emission wavelengths (Figure 2, Table 5).

The occurrence of peak A (λ_ex_/λ_em_ ytt80/375) is associated with the presence of biodegradable low-molecular-weight structural components and can be attributed to aromatic amino acids in protein-like systems [39,40,41] and/or phenolic compounds [41]. The B (λ_ex_/λ_em_ = 320/385) and C peaks (λ_ex_/λ_em_ = 370/430), in turn, reflect the fluorescence of hard-to-biodegrade terrestrial organic compounds of the fulvic-like and humic-like type with low and high molecular weight, respectively, and can be attributed to components derived from lignin and other degraded plant tissues [39,42,43]. The trend in fluorescence intensity changes in the studied humin is as follows: peak B > peak A > peak C. This observation indicates the predominance of unsaturated bond systems in the humin structure, such as aromatic structures with various types and numbers of substituents, capable of a high degree of conjugation. On the other hand, the relatively high intensity of peak A indicates a continuous “supply” of fresh material susceptible to microbiological degradation. Higher fluorescence efficiency was proven for the H10 rather than the H9 sample. This may be due to the smaller size of the molecule and/or different substitution in the aromatic ring. For example, the incorporation of electron-donating groups (hydroxyl, amino) increases the fluorescence efficiency, while the dominance of electron-withdrawing groups (carboxyl, carbonyl) is reflected in its decrease [35,36].

When interpreting fluorescence results, the B/A, C/B and HIX coefficients presented in Table 5 are often used. According to Albrecht et al. [39], changes in the B/A and C/B ratios may be related to the rate of the mineralization and humification of SOM processes of easily degradable compounds. Thus, a higher HIX coefficient in H9 molecules indicates more aromatic structures and/or an increase in the degree of conjugation of unsaturated aliphatic chains compared to H10. Moreover, lower HIX values are characteristic of organic matter that is more susceptible to microbiological degradation [44,45].

### 2.5. NMR Spectra

The carbon distribution found in the ^13^C CPMAS NMR spectra of humin fraction H9 and H10 (Figure 3) revealed a shared relevant incorporation of apolar alkyl (0–45 ppm) and aromatic (110–160 ppm) components, whose global relative amount accounted for 59.2% and 50.6% of total area for H9 and H10, in this order (Table 6). The most distinctive signals identified around 31–33 ppm and 127–128 ppm (Figure 3) are the signatures, respectively, of ordered or “crystalline” methylene segments of long-chain aliphatic molecules in lipids and biopolyesters of plant and microbial origin [46] and of un-substituted and C-substituted aromatic rings inherited from plant tissues or produced by the long-term secondary rearrangements of SOM. The presence of phenolic and lignin derivatives was highlighted by the O-bearing aryl C displayed at 145 and 151 ppm (Figure 3).

The band ranging from 45 to 60 ppm, marked by the peaks at 55/56 ppm, was derived from the layering of different components represented by either the methoxyl side groups of lignin units or the C-N bonds of peptidic moieties. The central wider resonances (60–110 ppm) are made up of the C nuclei of monomeric units in cellulose and hemicellulose of plant cell walls [47]. The former ones, distinguishable at lower chemical shifts (62/63 ppm), correspond to the out-of-plane C6 nuclei, followed by the intense sharp coalescence of C2, C3, and C5 at around 74 ppm (Figure 3). The noticeable shoulders refer to carbon 4 at 84 ppm, and the peak of di-O-alkyl C1 de-shielded to an upper chemical shift (105 ppm) denotes the typical β 1 → 4 glycosidic linkages of oligo and polymeric structures. The last peaks at 173 ppm are indicative of carboxyl/carbonyl functionalities arising from acids, amides and ester bonds [15].

The abundance of hydrophobic components is a peculiar characteristic of humin fractions formed through the preliminary selective preservation of recalcitrant aliphatic and aromatic molecules [11,17], whose low affinity for soil aqueous solution prompts a progressive tight interaction on soil minerals, thus fostering the incorporation and stabilisation of organic inputs [48,49]. The molecular features of H9 and H10 samples were outlined by the high values found for the A/OA and HB structural parameters (Table 6), which are related to the accumulation of stable humic components promoted by different SOM managements [50,51].

The most remarkable difference, shown by ^13^C CPMAS NMR spectra of humin fractions, is the larger magnitude of O-alkyl-C components found in the H10 sample from field plots with catch crops applied, underlined by the corresponding decrease in HB and A/OA values with respect to H9 (Table 6). This finding may imply that in a field with catch crop rotation, the regular biomass addition generates soil enrichment in labile carbon forms/pools promoted by the inherent stable core of humic compartments [52]. This trend is supported by the concurrent increase found in the LR of the H10 sample that, in combination with the steady yield of O-aryl-C, suggested the incorporation of biolabile C-N containing moieties from plant residues or microbial by-products [53]. Furthermore, the relative decrease found in the aromatic resonances (110–145 ppm) in H10 is in accordance with the decreased aromaticity reflected by its H/C ratio (Table 3) and fluorescence signals (Figure 2).

### 2.6. EPR Spectra

Radicals in both samples have the same structure, as the EPR g parameter remained unchanged (2.0029 ± 0.0001). A low parameter value is typical for radicals with an aromatic hydrocarbon or semiquinone structure [54]. Quantitative studies showed that humin H9 extracted from control soil samples contained more than twice as many radicals as humin H10 from fields where catch crops were used (4.5 × 10^18^ and 1.8 × 10^18^ respectively). This is consistent with the elemental composition, where sample H9 was characterised by a lower H/C ratio, indicating a higher content of aromatic groups than the H10 sample.

## 3. Material and Methods

### 3.1. Field Experiment Description and Soil Characteristics

The experiment was established in 1991 in Wrocław (southwestern Poland, 51.12° N; 17.14° E) at the Research and Education Station in Swojczyce of the Wrocław University of Environmental and Life Sciences. Climatic conditions of this area are characterised by annual rainfall of 576 mm and a mean annual temperature of +8.2° C. The experiment was established on a Dystric Cambisol (WRB soil classification) derived from loamy sand of alluvial origin, containing 77% sand, 18% silt and 5% clay. The plough layer (0–25 cm) had a slightly acidic reaction (pH_KCl_ 5.85–5.96), while total organic carbon content (TOC) and total nitrogen (TN) ranged from 10.2 to 12.8 g·kg^−1^ and 0.8 to 1.1 g·kg^−1^, respectively.

The aim of the long-term experiment was to determine the impact of the use of a catch crop (white mustard) on the yield of winter triticale grown in monoculture. Triticale was cultivated in 8 m wide strips in two variants: with and without catch crops. In each strip, three plots with an area of 20 m^2^ were separated as replications.

Each year, after harvesting the triticale, straw was left in the field and the soil was cultivated (discing); then, mustard was sown (at a dose of 20 kg/ha), and N fertilisation was applied (20 kg N/ha). After about a month, the catch crop mass was shallowly ploughed, and then, the soil was cultivated before sowing (ploughed to a depth of 20 cm) and fertilised (15 kg N/ha, 50 kg P_2_O_5_/ha; 80 kg K_2_O/ha). In the last days of September, winter triticale was sown (‘Rotondo’ variety). In the next growing season, in spring, fertilisation with N was applied at three phases of triticale development (60 kg N + 50 kg N + 40 kg N). At the turn of July/August, the triticale was harvested, while the straw was left in the field.

Samples were collected in 2022 from a layer of 0–20 cm A horizon of soil where catch crop was applied (sample H10) and that without catch crop (sample H9). Three plots of each variant were selected, and soil samples were taken from 10 points in each block plot. Each soil sample was dried at room temperature, then the roots and other plant remnants were removed. The material was ground and sieved to pass a 2-mm sieve.

### 3.2. Methods

#### 3.2.1. Fractional Composition of Humus Substances

Humic substances were extracted from the topsoil horizon using the procedure described by Swift [55] with some modifications. In the first step, the decalcitation of the soil sample was carried out using H_2_SO_4,_ to obtain a low-molecular-weight fraction, the so-called fulvic fraction (FF). In the supernatant, TOC content was determined (FF) using an Enviro TOC + N (Elementar; Langenselbold, Germany) analyser. In the next step, exhaustive alkaline extraction (with 0.1 M NaOH) was carried out until the supernatant was light in colour. This fraction was composed of humic acids and fulvic acids and named the total extractable carbon (TEC). The C concentration in the supernatant containing fulvic acids (FAs) and HA was determined. The HA fraction was precipitated by the acidification of the supernatant at approximately pH 2 of the alkaline extract, and solid HA was separated from FA by centrifugation. HA was then dissolved using hot 0.02 M NaOH, and the C concentration was measured in the entire volume of the solution (HA). Fulvic acid carbon (FA) was calculated using the formula FA = TEC – HA. The carbon of the non-hydrolysable humic substances, commonly referred to as the humin fraction, was calculated from the difference between TOC and sum of FF, HA and FA.

#### 3.2.2. Humin Fraction Isolation

Soil materials from the replicates were mixed to obtain an average sample from which the humin fraction was separated according to the method described in our previous publication [56]. In brief, humic and fulvic acids were extracted with NaOH. The remainder of the soil was digested with the HF-HCl mixture for removal of the mineral fraction. The material was then neutralised. It was purified by dialysis and freeze-dried.

#### 3.2.3. Elemental Composition of Humin Fraction

The bulk elemental composition of the humin was analysed on a CHNS Vario EL Cube analyser (Elementar; Langenselbold, Germany). The oxygen concentration was calculated from the mass balance (O% = 100 − C% − H% − N%). The content of elements is presented in ash free mass in atomic %. Table 2 presents the atomic ratios of the analysed humin samples.

#### 3.2.4. UV-Vis Properties of Humin

The UV-Vis analysis was caried out using a UV-VIS-NIR 770 spectrophotometer (Jasco, Tokyo, Japan). The UV–Vis spectra were recorded in the wavelength range from 200 to 700 nm. The optical path length was 1 cm. The humin samples were dissolved in a mixture of DMSO + 6% (*v*/*v*) H_2_SO_4_ (98% mass) solution to obtain a carbon concentration of 10 mg dm^−3^.

Firstly, the prepared solutions were subjected to ultrasound to achieve high sample homogeneity, and then, they were mixed on an orbital shaker for 24 h at a shaking speed of 160 rpm. After this time, the humin solutions were pre-filtered through 0.2 mm pore diameter filter paper. Before the analysis, humin solutions were filtered through a syringe filter with a pore size of 0.45 µm.

#### 3.2.5. Fluorescence Properties

Fluorescence analysis was caried out using an F 7000 fluorescence spectrophotometer (Hitachi, Chiyoda, Japan). Three-dimensional fluorescence spectra were scanned at emission wavelengths from 250 to 600 nm by changing the excitation wavelength from 200 to 550 nm and were analysed in the form of an excitation–emission matrix (EEM). The width of the excitation and emission slits was 10 nm, and the scanning speed was 1200 nm min^−1^. The fluorescence spectra were obtained for humin solutions prepared analogously to those for the UV-Vis.

#### 3.2.6. NMR

The 13C-CPMAS-NMR spectra of humin samples were recorded with a Bruker AVANCE 300 NMR spectrometer (Billerica, MA, USA) equipped with a 4 mm wide Bore MAS probe. The powdered samples were packed in 4 mm Zirconia rotors locked with Kel-F caps and spun at 10 kHz. About 28,000 scans were collected for each analysis over an acquisition time of 25 ms and a recycle delay of 2.0 s. The processing of acquired free induction decays (FIDs) was performed by 4 k zero-filling and line broadening of 200 Hz.

For the identification of the main organic compounds, the overall chemical shift range was split into six chemical shift regions: 0–45 ppm (alkyl-C), 45–60 ppm (CH_3_Ol-C and N- -C bonds), 60–110 ppm (O-alkyl-C), 110–145 ppm (aryl-C), 145–160 ppm (O-aryl-C), and 160–190 ppm (carboxyl-C). The ponderal contribution of single organic classes has been evaluated as relative intensity with respect to total spectral area. The molecular properties of organic substrates can be summarised by a dimensionless structural index determined by the combination of specific functional groups in NMR spectra [50,53].

The alkyl ratio estimates the relative content of alkyl-C as compared to O-alkyl-C: A/OA = (0–45 ppm)/(60–110 ppm)

The Hydrophobic Index (HB) relates the amounts of hydrophobic apolar C functionalities to those of potentially more hydrophilic polar functional groups: HB = S[(0–45 ppm) + (45–60 ppm)/2 + (110–160 ppm)]/S[(45–60 ppm)/2 + (60–110 ppm) + (160–190 ppm)].

The lignin ratio (LR) was obtained from the C distribution found in the lCH3O-C C-N region over the content of O-Aryl-C compounds: LR = (45–60 ppm)/(145–160 ppm)

The values of A/OA and HB parameters have been employed to assess the dynamics of soil organic matter fractions. The LR is a useful indicator to assign the NMR signals in the 45,060 ppm interval to either lignin phenolic compounds (lower LR) or to peptidic derivatives (larger LR).

#### 3.2.7. EPR

The X-band EPR spectra were recorded at room temperature using a Bruker Elexsys E500 spectrometer with a double rectangular cavity resonator specifically designed for quantitative measurements. The humic acid standards of Pahokee peat (1S103H) and the Leonardite (1S104H) extracted and distributed by the International Humic Substances Society (IHSS), in addition to the Bruker alanine pill, were used as quantitative standards. To calculate the radical concentration, a double integration of the signals (samples and standards) with the use of the WINEPR V2.22Rev 12 programme by Bruker was performed. As the radicals in humic substances are organic, results were recalculated to ash-free mass.

#### 3.2.8. Statistical Analysis

Statistical analysis was performed using an analysis of variance and the mean differences were compared by a post hoc test at a *p* level of <0.05, according to Tukey’s HSD. Pearson’s correlation analysis was performed to investigate the multivariate relationships between the examined humus fractions. All analyses were carried out using the data analysis programme Statistica 13.3 (TIBCO Software Inc. Santa Clara, CA, USA).

## 4. Conclusions

Our studies have shown that the more than 30-year application of catch crops did not cause significant changes in the soil organic carbon content, but it significantly affected the transformation routes of organic matter. Catch crop cultivation favoured the formation of highly reactive humus substances, such as low-molecular-weight fractions, and also resulted in a higher share of humic acids. On the other hand, this kind of cultivation caused a decrease in the share of the humin fraction. The higher H/C and O/C atomic ratios of humin molecules originating from soil where catch crops were applied pointed out that their structure is more abundant in oxygen-containing moieties and less aromatic. The UV-Vis, fluorescence and ^13^C-CPMAS-NMR investigations confirmed a relatively higher share of oxygen-containing functional groups in humin isolated from the soil with catch crop rotation, also corroborating its greater aliphatic nature. The findings of spectroscopic studies of humin originated from the plots with catch crops suggested its lower degree of condensation of aromatic structures, smaller molecular size and/or diverse substitution in the aromatic ring.

Our research indicates that the organic residues from catch crops quickly undergo the decay process and transform mainly into highly reactive humic substances, which improve soil health, while mineral fertilisation alone without catch crops favours the stabilisation and sequestration of carbon.

## Figures and Tables

**Figure 1 molecules-29-05049-f001:**
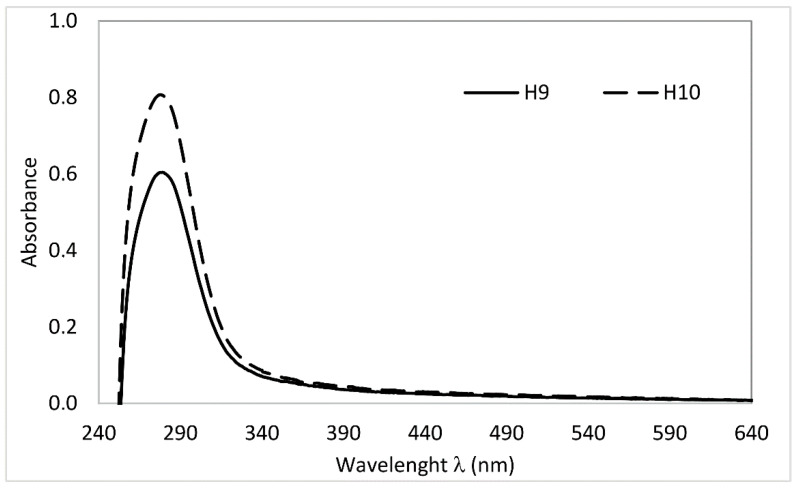
UV-Vis spectra of studied humin.

**Figure 2 molecules-29-05049-f002:**
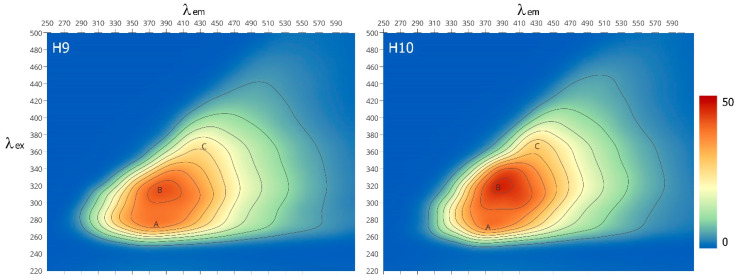
EEM spectra of analysed humin. Letters A, B and C indicate areas of characteristic fluorophore structures.

**Figure 3 molecules-29-05049-f003:**
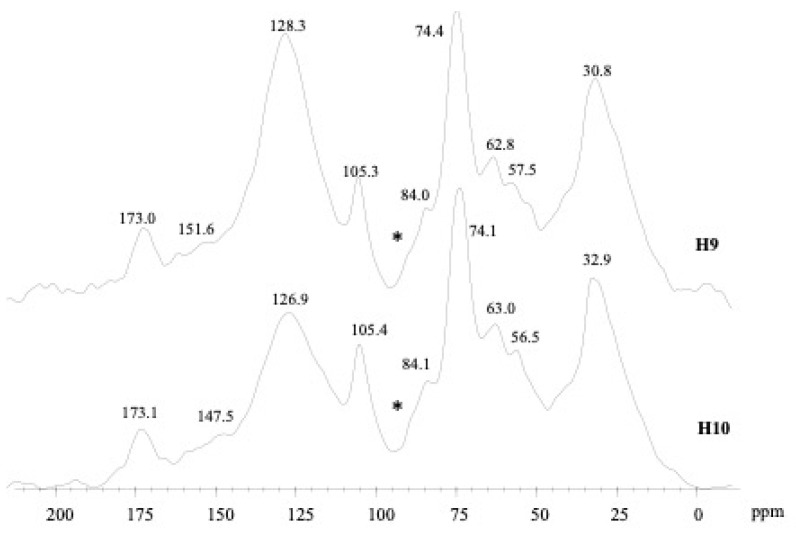
^13^C-CPMAS NMR spectra of humin samples (the asterisks indicate the approximated position of side bands of carbonyl carbons).

**Table 1 molecules-29-05049-t001:** Fractional composition of humus fractions.

Sample No	TOC g kg^−1^	FF	TEC	HA	FA	HUM	HA/FA
		C% in TOC					
H9	12.3 ^a^	9.6 ^a^	53.6 ^a^	29.7 ^a^	23.9 ^a^	36.8 ^a^	1.2 ^a^
H10	11.5 ^a^	11.3 ^b^	63.4 ^b^	37.1 ^b^	26.3 ^a^	25.3 ^b^	1.4 ^a^

^a,b^ means followed by the same letter are not significantly different at *p* < 0.05. TOC—total organic carbon content; FF—low-molecular-weight fraction; TEC—total extractable carbon; HA—humic acids; FA—fulvic acids; HUM—humin.

**Table 2 molecules-29-05049-t002:** Correlation coefficients for the humus fractions in the soil without catch crops.

	TOC	FF	TEC	HA	FA	HUM	HA/FA
TOC	1.000	−0.999 *	0.650	0.924	0.492	−0.621	−0.213
FF		1.000	−0.655	−0.927	−0.498	0.627	0.220
TEC			1.000	0.891	0.981	−0.999 *	−0.881
HA				1.000	0.787	−0.873	−0.569
FA					1.000	−0.988	−0.955
HUM						1.000	0.898
HA/FA							1.000

Values of the coefficient marked with * have significant correlation.

**Table 3 molecules-29-05049-t003:** Elemental composition of analysed humin fraction.

Sample No	Ash %	C	H	O	N	S	H/C	O/C	N/C
		Atomic % in Ash Free Sample							
H9	18.18	50.75	33.78	14.11	1.25	0.11	0.67	0.28	0.04
H10	9.76	44.54	34.64	19.34	1.40	0.08	0.78	0.44	0.09

**Table 4 molecules-29-05049-t004:** Spectral ratios calculated for the tested humin.

Sample No	E_465_:E_665_	E_280_:E_665_	E_280_:E_465_	E_280_:E_400_
H9	2.88	82.16	28.32	18.27
H10	4.03	126.88	31.16	20.54

**Table 5 molecules-29-05049-t005:** The intensity of fluorescence maxima for peaks A, B and C, and selected indexes.

Sample No	Peak A	Peak B	Peak C	B/A	C/B	HIX
H9	40.6	43.9	24.6	1.08	0.56	1.38
H10	42.4	48.8	28.9	1.15	0.59	1.18

**Table 6 molecules-29-05049-t006:** C distribution (%) along chemical shift regions (ppm) in ^13^C CPMAS NMR spectra and associated structural indexes of humin samples.

Sample No	C=O 190-160	O-Aryl-C 160-145	Aryl-C 145-110	O-Alkyl-C 110-60	CH_3_O/C-N 60-45	Alkyl-C 45-0	A/OA	HB	LR
H9	5.0	4.5	29.6	29.8	5.9	25.1	0.8	1.6	1.3
H10	4.9	4.6	22.1	36.3	8.1	23.9	0.6	1.2	1.8

## Data Availability

The data presented in this manuscript are available from the corresponding author on request.

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
