# Peer review of "Changes in Soil Humin Macromolecular Structure Resulting from Long-Term Catch Cropping"

_molecules, 2024, doi:10.3390/molecules29215049_

Round 1
Reviewer 1 Report
Comments and Suggestions for Authors
The paper entitled: „Changes in the humin macromolecular structure resulting from the long-term catch-crop application” is very interesting, written in a reader-friendly and clear manner. Results and discussion is very correct. The topic and the scope of research comply with the journal's profile.
The paper addresses a very important issue – the properties of humins. Humins sometimes constitute over 60% of organic carbon (TOC). Despite such a large share of humins in organic matter, humins are a fraction of organic matter whose properties (structure) are rarely studied. The main reason is the difficulty associated with their isolation. The content of humins in soil can have a significant impact on carbon sequestration. The authors of the paper not only determine the properties of humins, but also the impact of 30-year field experiment on triticale cultivated with and with-out catch crops on their structural properties. However, before publishing, the paper requires a few changes and some supplementary information which, to my belief, will enhance the content-wise value of the paper.
1. The authors of the paper describe the fractionation process in detail, but they did not provide information on how they prepared the humin samples for analysis. Please provide additional information.
2. The authors of the work in Table 1 provide the significance of differences between variants < 0.05 and in Table 2. the values of correlation coefficients. Please supplement the subsection "Methods" with the statistical methods that were used. As it results from the data presented in Table 2, only two significant correlation dependencies were found - are you sure?
3. Line 92-93 I would suggest not using the phrase "highly reactive humus substances" in reference to humic acids, if at all, they are more reactive than humins.
4. Line 349 is LigR should be LR
5. Line 345-347 please check if the given formula (HB) is correct, what does the letter S mean in the formula.
6. Check and correct the list of literature (errors happen)
Author Response
Authors would like to express their gratitude to the reviewer for insightful and kind suggestions, which were considered constructive and supportive. All of them were addressed, which in our opinion significantly improved the quality of the manuscript. The text has been revised according to comments and suggestions of three reviewers. Below are all comments of the reviewer followed by authors responses marked in blue.
- The authors of the paper describe the fractionation process in detail, but they did not provide information on how they prepared the humin samples for analysis. Please provide additional information.
The following description has been added:
3.2.2. Humin fraction isolation
Soil materials from the replicates were mixed to obtain an average sample from which the humin fraction was separated according to the method described in our previous publication [XX]. Briefly, humic and fulvic acids were extracted with NaOH, then the remaining soil was digested with the HF-HCl mixture to remove the mineral fraction. Finally, the material was neutralised, purified by dialysis, and freeze-dried.
- The authors of the work in Table 1 provide the significance of differences between variants < 0.05 and in Table 2. the values of correlation coefficients. Please supplement the subsection "Methods" with the statistical methods that were used. As it results from the data presented in Table 2, only two significant correlation dependencies were found - are you sure?
The following description has been added:
3.2.8. Statistical analysis
Statistical analysis was performed using analysis of variance and the mean differences were compared by post hoc test at a p level of <0.05, according to Tukey’s HSD. Pearson’s correlation analysis was performed to investigate the multivariate relationships between the examined humus fractions. All analyses were carried out using the data analysis programme Statistica version 13 (TIBCO).
As it results from the data presented in Table 2, only two significant correlation dependencies were found - are you sure?
Yes, only two significant correlation coefficients have been found.
- Line 92-93 I would suggest not using the phrase "highly reactive humus substances" in reference to humic acids, if at all, they are more reactive than humins.
The sentence has been corrected as follows:
First of all, the soil cultivated in this way (H10) showed a significantly higher amount of the low-molecular-weight fraction (FF) and humic acids (HA).
- Line 349 is LigR should be LR
LigR has been replaced with LR
- Line 345-347 please check if the given formula (HB) is correct, what does the letter S mean in the formula.
The letter S stand for summation;so it has been edited with Symbol typeface, as follows:
HB=S[(0-45ppm)+(45-60ppm)/2+(110-160ppm)]/S[(45-60ppm)/2+(60-110ppm)+(160-190ppm)].
- Check and correct the list of literature (errors happen)
The list of literature has been checked and corrected.
Reviewer 2 Report
Comments and Suggestions for Authors
In this work, the authors used several complementary techniques to assess the effect of long-term catch cropping on the structural properties of humin, considered the most recalcitrant fraction of soil organic matter. Soil samples from a long-term experimental field (over a 30-year) on triticale cultivated with and without catch crops were analysed for total organic carbon content and fractional composition of humic substances. It is found that catch crop farming enhanced the formation of highly reactive humus substances and humic acids while decreasing the amount of the humin fraction. The results indicated that organic residues from catch crops quickly undergo decay processes that can potentially improve soil health, while mineral fertilization alone without catch crops favours the stabilization and sequestration of carbon.
This work is very interesting and well-written with a clear structure. The topic is well-suited for the journal. Small duplication is detected (32% iThenticate analysis), most correspond with the M&M section. To the best of my knowledge (Clarivate WoK search), this work has not been previously published.
I didn't find any major issues, so I'm happy to recommend it for publication after minor changes.
Below are some observations and recommendations that I hope may help ameliorate this MS.
The title can be shortened to:
Changes in soil humin macromolecular structure resulting from long-term catch-cropping.
Avoid keywords appearing in the title
Abstract; write the last paragraph more prudently i.e.” Under the conditions of our field experiment, the results indicate ….”
Line 53-54. Please revise the sentence I do not know what the authors mean by “the circulation of ingredients in nature”.
Line 58. In a first appearance, please declare the meaning of the acronym (HS)
Line 82-83. Please revise, humin may be a fraction relevant for C sequestration, but is believed to be a low reactive form of C and I am not sure of its importance in safeguarding soil fertility.
Line 145. Check for spelling mistakes.
Fig 3. Include NMR spin side bands labelled with an asterisk.
In M&M. Please describe in detail the soil sample procedure.
Comments on the Quality of English Language
Several flaws are detected in the English language and a revision is recommended.
Author Response
Reviewer 2
Authors would like to express their gratitude to the reviewer for insightful and kind suggestions, which were considered constructive and supportive. All of them were addressed, which in our opinion significantly improved the quality of the manuscript. The text has been revised according to comments and suggestions of three reviewers. Below are all comments of the reviewer followed by authors responses marked in blue.
The title can be shortened to: Changes in soil humin macromolecular structure resulting from long-term catch-cropping.
The title has been shortened accordingly
Avoid keywords appearing in the title
Keywords appearing in the title have been deleted
Abstract; write the last paragraph more prudently i.e.” Under the conditions of our field experiment, the results indicate ….”
Thank you for suggestion. The sentence has been changed for: “Under the conditions of our field experiment, the results indicated that organic residues from catch crops quickly undergo the decay process and are transformed mainly into highly reactive humus substances, which can potentially improve soil health, while mineral fertilization alone without catch crops favours the stabilization and sequestration of carbon”
Line 53-54. Please revise the sentence I do not know what the authors mean by “the circulation of ingredients in nature”.
The word “ingredients” has been replaced with “elements”
Line 58. In a first appearance, please declare the meaning of the acronym (HS)
The acronym (HS) has been replaced with “humic substances”, because the term “humic substances” does not reappear throughout the document
Line 82-83. Please revise, humin may be a fraction relevant for C sequestration, but is believed to be a low reactive form of C and I am not sure of its importance in safeguarding soil fertility.
The sentence has been revised as follows: “There are no studies in the literature on the influence of this type of cultivation on the quality of organic matter, in particular on the properties of the humin fraction, which is a fraction relevant for C sequestration”.
Line 145. Check for spelling mistakes.
The sentence has been corrected as follows;
Furthermore, the higher value of the E280:E400 coefficient observed for H10 extracted from catch crop applied soil points to its relatively higher proportion of oxygen-containing functional groups, including carboxyl groups [38]. This confirms its greater aliphatic nature, as indicated by the results of elemental composition analysis (Table 3).
Fig 3. Include NMR spin side bands labelled with an asterisk.
The Figure 3 has been updated, and corresponding asterisks have been included
In M&M. Please describe in detail the soil sample procedure.
The following description has been added:
Samples were collected in 2022 from a layer of 0-20 cm A horizon of soil where catch crop was applied (sample H10) and that without catch crop (sample H9). Three plots of each variant were selected, and soil samples were collected from 10 points in each block plot. Each soil sample was dried at room temperature, then the roots and other plant remnants were removed. The material was ground and sieved to pass a 2-mm sieve.
Several flaws are detected in the English language and a revision is recommended.
English has been revised